# Neural Predictors of the Antidepressant Placebo Response

**DOI:** 10.3390/ph12040158

**Published:** 2019-10-19

**Authors:** Danielle Rette, Erin McDonald, Dan V. Iosifescu, Katherine A. Collins

**Affiliations:** 1Nathan Kline Institute for Psychiatric Research, Orangeburg, NY 10962, USA; Danielle.Rette@nki.rfmh.org (D.R.); Dan.Iosifescu@NKI.rfmh.org (D.V.I.); 2Department of Psychiatry, New York University School of Medicine, New York, NY 10016, USA; 3Department of Psychiatry, Icahn School of Medicine at Mount Sinai, New York, NY 10029, USA

**Keywords:** placebo response, antidepressant, neural predictors

## Abstract

The antidepressant placebo response remains a barrier to the development of novel therapies for depression, despite decades of efforts to identify and methodologically address its clinical correlates. This manuscript reviews recent neuroimaging studies that aim to identify the neural signature of antidepressant placebo response. Data captured in clinical trials have primarily focused on antidepressant efficacy or predicting antidepressant response and have reliably implicated the rostral anterior cingulate cortex (rACC) in antidepressant placebo response, but also in medication response. Imaging and electroencephalography (EEG) experiments specifically interrogating the mechanism of antidepressant placebo response, while few, suggest the reward network, including opiate neurotransmission, is also involved. Therefore, while the rACC is likely involved in the antidepressant placebo response, its observation in isolation is unlikely to prospectively distinguish antidepressant placebo from medication responders. Instead, future studies of antidepressant placebo response should probe the reward network as a whole and incorporate sophisticated computational analytical approaches.

## 1. Introduction

Depression is the primary cause of disability worldwide, affecting an estimated 300 million individuals and their families [1]. Traditional monoaminergic antidepressants can take weeks to relieve symptoms and precipitate complete remission in only one-third of treated patients [2]. Recent FDA-approval of medications for treatment-resistant (esketamine) and postpartum (brexanolone) depression with unique mechanisms of action offers new hope for patients and scientists alike. Still, esketamine is a controlled substance, and both therapies must be administered in medical facilities enrolled in a risk evaluation and mitigation strategy program, increasing cost and reducing accessibility. There is a clear unmet need for novel treatments, particularly agents that are easily and safely distributed and administered. 

To ensure the regulatory approval of a novel antidepressant, studies must demonstrate not only that an investigational agent reduces depression severity, but also that its efficacy is statistically greater than that of a placebo (or sham, if testing a device). Robust placebo response rates in clinical trials [3,4] therefore reduce the likelihood of demonstrating the relative efficacy of an investigational agent [5] and have contributed to the demise of several promising potential antidepressants. Here, the pharmaceutical field has identified reproducible clinical correlates of antidepressant placebo response (e.g., disease severity, comorbid anxiety) and modified inclusion and exclusion criteria accordingly [6], with the aim of preventing likely placebo responders from entering the pool of randomized clinical trial participants. Methodologists, in addition, have innovated and tested a multitude of increasingly complex trial designs [6,7,8,9] and strategies of assessment [10,11,12,13,14], with the aim of reducing placebo response rates. Unfortunately, efforts to date have largely been unsuccessful.

More recently, neuroimagers have joined the fray, aiming to identify the neural signature of antidepressant placebo response. This paper will review neuroimaging studies of the antidepressant placebo response and discuss the implications of the findings to date. 

## 2. Imaging Data Captured in Clinical Trials Primarily Focused on Antidepressant Efficacy or Predicting Antidepressant Response

Most studies purporting to explore imaging correlates of antidepressant placebo response have exploited data captured in the context of traditional placebo-controlled antidepressant trials, as summarized in Table 1. 

Mayberg and colleagues [15] have examined regional cerebral glucose metabolism in 17 inpatient men with major depressive disorder (MDD). The patients were treated on an inpatient research unit for 6 weeks and had an average score of 22 (SD = 5) on the seventeen-item Hamilton Depression Rating Scale (HAMD-17). Using a double-blind study design, patients were assigned to a fixed dose of either fluoxetine (20 mg/day) or a placebo. Glucose metabolism was measured using fluorodeoxyglucose (FDG) and positron emission tomography (PET) before and after 1 and 6 weeks of fluoxetine or placebo administration. Symptom remission was seen in 8 of the 15 study completers. Four of the 8 responders were treated with a placebo. The responders were defined as having at least a 50% decrease in HAMD-17 scores relative to baseline scores. They found that the placebo response was associated with significant increases and decreases in glucose metabolism in both the neocortical and limbic-paralimbic regions. Areas evincing increased metabolism included the dorsolateral prefrontal cortex, premotor cortex, inferior parietal cortex, posterior insula, and posterior cingulate. Areas evincing decreased metabolism included the rostral anterior cingulate cortex (rACC), hypothalamus, thalamus, supplementary sensory area, and para-hippocampus. While comparable brain changes were seen in both the drug and placebo responders in the specific cortical and paralimbic regions, active fluoxetine treatment was associated with additional changes in the brainstem, striatum, anterior insula, and hippocampus. 

Leuchter and colleagues [16,17] aimed to determine the neurophysiological, symptomatic, and cognitive characteristics of individuals likely to respond to placebos in clinical trials for MDD. A total of 51 subjects completed one of two randomized, double-blind clinical trials, comparing treatment with a medication or placebo (fluoxetine/placebo n = 24, venlafaxine/placebo n = 27) for 8 weeks. Changes in depression severity were tracked with the HAMD-17 scale. Both studies also included a one-week single-blinded placebo lead-in phase. Patients completed a series of cognitive assessments prior to treatment administration. Electroencephalography (EEG) data, in addition, was recorded from 35 scalp electrodes at baseline, after the placebo lead-in, and 2, 4, and 8 weeks after randomization. EEG analysis included calculation of concordance, a measure with a “moderately strong” association with cerebral blood flow, as assessed by O^15^ PET. At week eight, investigators characterized participants as medication responders (n = 13), medication non-responders (n = 12), placebo responders (n = 10), and placebo non-responders (n = 16). In an initial analysis in which subsets of individual electrodes were grouped into regional measures, no group differences in baseline EEG values were documented. Instead, placebo responders evinced an increase in prefrontal (FP1 and FP2) theta concordance that, at week eight, statistically distinguished them from all other groups. However, when the data from all 35 individual electrode sites were included in statistical models, baseline theta (4–8 Hz) concordance measured at two frontocentral sites (AF1 and AF2) was significantly lower in the placebo responders compared to all other subjects, and medication responders in particular. In addition, at baseline, placebo responders outperformed the comparison groups on the digit symbol test (measuring information processing speed) and reported significantly less late insomnia than all other groups. Logistic regressions showed that a model including frontocentral concordance, digit symbol test scores, and late insomnia severity classified nearly 98% percent of subjects (7 of 8 or 87.5% of placebo responders and 34 of 34 or 100% of other) as either placebo responders or other (medication responders, medication non-responders, placebo non-responders). In a separate model, frontocentral concordance was the only variable that distinguished placebo responders from medication responders specifically, correctly classifying 75% of included subjects into one of these groups. 

In a subsequent analysis of data gathered in the same clinical trials, however, including additional subjects and employing a different method of EEG analysis, Korb and colleagues [18] assessed whether pretreatment theta current density in the rACC and medial orbitofrontal cortex (mOFC) differentiated placebo and medication responders from non-responders. They included the EEG data collected from 72 subjects (fluoxetine n = 13, venlafaxine n = 24, placebo n = 35) and applied low resolution electromagnetic tomography (LORETA) to compute theta-band current density for each subject. The authors observed that eventual medication responders displayed higher baseline theta current density in both the rACC and mOFC. However, placebo responders showed no significant differences in baseline theta current density compared to placebo non-responders in either region of interest.

More recently, three papers have analyzed the Establishing Moderators and Biosignatures of Antidepressant Response to Clinical Care (EMBARC) study in an effort to identify neural predictors of the antidepressant placebo response. The EMBARC study includes participants with recurrent and nonpsychotic MDD who were randomized to 8 weeks of sertraline or placebo. EEG data were recorded at baseline and following 1 week of treatment. Depression severity was measured using the HAMD-17 scale. Trivedi, South, and Jha et al. [19] have investigated clinical and EEG markers of placebo response by evaluating data from the subset of participants (n = 141) who were assigned to the placebo arm of the EMBARC trial. The authors used advanced variable selection methods to explore 283 variables suspected to predict the degree of symptom change and the probability of remission and response. They identified predictors of placebo response, including lower baseline depression severity, younger age, absence of melancholic features, absence of a history of physical abuse, less anxious arousal, less neuroticism, and higher average pretreatment theta current density in the rACC. 

Pizzagalli, Webb, and Dillon et al. [20] have analyzed EEG data from all study participants (n = 248), including those randomized to both drug and placebo treatment, showing that baseline rACC theta activity was negatively associated with HAMD-17 scores at week 8, therefore predicting a greater improvement in depression severity. Furthermore, for every standard deviation increase in rACC theta activity, there was a 1.5-point decrease in week 8 HAMD-17 scores. The association between rACC theta activity and better outcomes was not impacted by treatment group. Therefore, participants with higher baseline and week 1 rACC theta activity had better outcomes regardless of antidepressant or placebo assignment.

Finally, Whitton and colleagues [21] used the same EEG data to evaluate whether rACC connectivity with the posterior cingulate cortex (PCC), left dorsolateral prefrontal cortex (lDLPFC), or the right anterior insula (rAI) contributed to depression symptom improvement. Their analysis included data from a subset of 238 subjects. Regions of interest other than the rACC were selected because of their roles as hubs in the default mode network (PCC), frontoparietal network (lDLPFC), and salience network (rAI). Results showed that stronger baseline theta-band rACC-rAI connectivity predicted greater depression symptom improvement after 8 weeks for both groups (drug and placebo treatment). Furthermore, earlier increases in theta-band rACC-rAI connectivity predicted a higher chance of remission at 8 weeks. 

## 3. Imaging Experiments Specifically Interrogating Mechanism of Antidepressant Placebo Response

Peciña [22] and Sikora [23] and colleagues conducted a comprehensive and complex neuroimaging investigation, directly probing antidepressant placebo response, enrolling participants diagnosed with MDD who were not taking antidepressants. 

The study began with a 2-week single-blinded randomized controlled cross-over phase. Participants were initially randomized to 1-week of treatment with either an “active” placebo, wherein study staff explicitly instructed them to expect a fast-acting antidepressant, or an “inactive” placebo, wherein study staff explicitly instructed them to expect an inactive control. After a wash-out period, participants crossed over into the alternate condition. After each week of placebo, participants underwent functional magnetic resonance imaging (fMRI, n = 29) to collect resting-state data and PET scanning (n = 35), utilizing the µ-opioid receptor (MOR)-selective radiotracer [^11^C] carfentanil to assess MOR neurotransmission. The PET session after 1 week of the “active” placebo included intravenous (IV) injections of additional doses of “active” placebo at timed intervals. That is, 1 ml saline injections lasting 15 s were administered every 4 min for a 20-min interval. At the start of each “active placebo” injection, investigators played a computer-generated human voice recording to alert subjects to the administration and displayed a second-by-second countdown for the duration.

To quantify “sustained placebo response”, the authors calculated the difference in Quick Inventory of Depressive Symptomatology-Self Report (QIDS-SR) scale score changes observed in the active and inactive placebo treatment periods treatments [(QIDS-SR16 pre-post) “active” placebo—(QIDS-SR16 pre-post) “inactive” placebo]. Using this rubric, positive values indicated greater reductions in depression severity from the oral “active” placebo versus the “inactive” placebo and were used to classify participants as placebo responders (n = 14) or non-responders (n = 21). In a second phase, participants received 10 weeks of open-label treatment with an antidepressant. 

As predicted, a week of “active” placebo treatment was associated with a significant reduction in depression symptoms when compared with a week of “inactive” placebo. In addition, acute IV placebo administration was associated with significant reductions in patients’ depression severity ratings. Analysis of PET data [22] revealed significant activation of MOR neurotransmission after the IV placebo was localized in the nucleus accumbens (NAc), rACC, and amygdala. Improvement evinced after one week of treatment with the “active” oral placebo group was positively associated with IV placebo induced opioid release in the, NAc, rACC, amygdala and midline thalamus. Higher baseline MOR binding potential in the NAc was associated with higher depression symptomology and subsequent antidepressant, but not placebo responsiveness. However, IV placebo induced MOR system activation in the rACC, NAc, amygdala, and thalamus predicted 43% of variance to response to antidepressants after 10 weeks. 

fMRI data from the same study implicated the rACC, as well as the salience network, in antidepressant placebo response [23]. The salience network (Figure 1), with hubs in the bilateral anterior insula cortices and the dorsal anterior cingulate cortex (dACC), also includes the amygdala, ventral striatum, ventral tegmental area/substantia nigra, dorsomedial thalamus, hypothalamus, and periaqueductal gray [24]. The resting state functional connectivity of the rACC within the salience network, as assessed after one week of “inactive” oral placebo treatment, was positively correlated with a reduction in depression severity in response to both the oral “active” placebo and, subsequently, ten weeks of open-label treatment. The degree to which rACC resting-state functional connectivity within the salience network was reduced after the “active” placebo, as compared to the “inactive” oral placebo, was also significantly associated with placebo response. Multivariate relevance vector regression, a machine learning tool, showed that the resting state functional connectivity of the entire salience network was significantly predictive of placebo responses but not drug responses. However, a validation of this model will require a replication of this result.

Peciña [25] lead another direct interrogation of antidepressant placebo response using a sham neurofeedback model. During fMRI scanning, twenty-four depressed subjects completed a within-subject trial-by-trial manipulation of two components of placebo antidepressant effect, namely, expectation of mood improvement and its reinforcement. Participants’ expectation of mood improvement was manipulated by displaying ten-second cues instructing either that an infusion of a “fast-acting antidepressant” was about to begin or that “no infusion” would be administered. Saline was then infused. Subsequently, subjects viewed sham neurofeedback, showing either small increases, large increases, small decreases, or large decreases in brain activity (reinforcement). An instructional video played prior to the experiment explicitly instructed participants that increases in brain activity were indicative of drug efficacy and predictive of subsequent mood improvement while decreases in brain activity should be interpreted as normal responses in the absence of drug treatment and could trigger mood worsening. The video also informed participants that the magnitude of change in brain activity would correspond to the degree of the drug’s efficacy. After each infusion, subjects rated their expected, and subsequent experience of, mood change on a scale ranging from −3 (much worse) to +3 (much better). Subjects’ depressive symptoms were assessed using the Montgomery-Åsberg Depression Rating Scale (MADRS) and the HAMD-17. Blood samples were also collected from seventeen subjects one hour before and immediately following the scan. 

As expected, patients expected greater mood improvement when anticipating an infusion (versus no infusion). Greater mood improvement during the infusion was associated with greater expectation of clinical benefit, the infusion condition compared to the non-infusion condition, and the positive sham compared to the negative sham. They found that positive neurofeedback of a greater magnitude (high > low) recruited greater blood-oxygen-level dependent (BOLD) responses in the bilateral ventrolateral and dorsolateral prefrontal cortical regions (VLPFC/DLPFC). The rostral/dorsal ACC and ventral striatal clusters did not survive correction for multiple comparisons. Furthermore, greater increases in β-endorphin plasma levels were associated with a significantly greater expectation of benefit and greater subjective mood improvement with positive neurofeedback.

## 4. Discussion

### 4.1. Endogenous Opioids and the Antidepressant Placebo Response

Two studies unequivocally demonstrate that endogenous opioid neurotransmission is involved in antidepressant placebo response [22,23]. Greater increases in β-endorphin plasma levels during an fMRI session, when incorporating treatment with an IV placebo, were linked to significantly greater expectation of clinical benefit and greater subjective mood improvement with sham neurofeedback. In a separate PET study, extemporaneous IV placebo delivery also precipitated µ-opioid release in the rACC, NAc, and amygdala. Reduction in depression symptoms with an active placebo was also associated with µ-opioid release in the rACC, NAc, amygdala, and midline thalamus. 

### 4.2. The Reward Network and Antidepressant Placebo Response

Opioid neurotransmission is one of the two primary chemical effectors of the experience of reward [26,27]. It is therefore consistent that all the brain regions that have been repeatedly implicated in antidepressant placebo response are also known participants in the “reward network” (Figure 1). The reward network includes the entire salience network, as well as the vmPFC, including the rACC, orbitofrontal, and lateral prefrontal cortical regions, and is known to encode both anticipation and receipt of reward [28,29]. 

This connection between placebo and reward may be understood in the context of the expectation of clinical benefit. The expectation of clinical benefit is the most reliable predictor of [30,31], and has been theorized to underlie [32] antidepressant placebo response. The more likely a participant believes she is to receive an active versus an inactive treatment, the more likely she is to report a subsequent reduction in depression severity. Further, expectation of clinical benefit can be conceptualized as the anticipation of a reward [33]. Therefore, it follows that neural systems engaged in predicting reward would also mediate antidepressant placebo response. In fact, citing converging results across conditions (e.g., pain and Parkinson’s), De La Fuentez [34] has proposed the “placebo-reward hypothesis”, which predicts that any placebo response, regardless of the medical condition, should be associated with the activation of reward circuitry in the brain.

### 4.3. Rostral Anterior Cingulate Cortex and Antidepressant Placebo Response

The rACC is the brain region most frequently identified as a biomarker of antidepressant placebo response. In four distinct patient cohorts, researchers have found that heightened resting baseline rACC theta current density [19,20] and rACC connectivity with the salience network [23] or its hub, the rAI, [21] were associated with, or even predicted, the antidepressant placebo response. In addition, reductions in rACC activity [15] or salience network connectivity [21,23] accompanied the antidepressant placebo response.

However, in all investigations, rACC metrics were also predictive of medication response. Therefore, while these findings could have multiple clinical implications (See [19,20] for thorough discussion), rACC observation is unlikely to advance the capacity to prospectively identify antidepressant placebo responders. This should not be surprising, since in most of the studies linking the rACC to both placebo and treatment response, the rACC was isolated as an a priori region of interest, given its established role in generalized treatment response, rather than emerging as a key structure in agnostic whole-brain analyses [18,19,20,21,23]. 

### 4.4. Consistent Overlap in Neural Substrates of Antidepressant Placebo and Medication Response

In addition to the rACC, all other individual brain regions that are now known to mediate antidepressant placebo response have also been implicated in medication response. While the overlap in brain regions implicated in response to antidepressant placebo and active antidepressant treatments is hardly surprising, the following question arises: Is it even possible to prospectively discriminate patients capable of responding to placebos from those only capable of responding to explicitly biologically active interventions? Perhaps there is no baseline biomarker of potential antidepressant placebo response or anything biologically unique about placebo responders. In this case, it may be most efficient to abandon efforts to prospectively exclude patients who might respond to placebos and instead utilize relapse response prevention studies to demonstrate a potential therapy’s superiority to placebos. Such trials are more likely to demonstrate superior efficacy of investigational agents [35]. Further, there is evidence that, while all brain changes observed with placebo treatment are also observed with active treatment, additional novel changes are associated with active treatment. Mayberg [15], for instance, reported that fluoxetine did precipitate unique changes in striatal and hippocampus metabolism after six weeks, and Pecina [22] found evidence that that MOR binding potential in the NAc at baseline predicted only response to medication. Hence, it is possible to prove that investigational agents incite different and potentially more crucial brain changes than placebos.

### 4.5. Salience/Reward Network Functional Connectivity and Other Tools for Future Brain-Based Prospective Identification of Likely Placebo Responders

Despite the obvious challenges, the data do offer hope that the prospective identification of potential placebo responders (and the subsequent reduction in antidepressant placebo response rates in clinical trials) is still possible. For instance, in the sample of patients studied by Sikora and peers [23], the application of machine learning revealed that functional connectivity of the salience network in its entirety did predict placebo response but not medication response. Efforts to replicate this finding are clearly warranted, as are efforts that combine clinical and imaging variables in models such as those described by Trivedi, South, and Jha [19].

Future brain imaging investigations may benefit from the use of similarly sophisticated and agnostic statistical strategies paired with computational analyses, rather than focused region-of-interest based analyses, which may have obscured key correlates in the past. Given the theoretical and data-based support for the placebo-reward hypothesis, it may also behoove researchers to probe not only brain regions in the salience network, but the entire (larger and all-encompassing) reward network. In a similar vein, examining the role of dopamine neurotransmission may prove fruitful, especially in the context of a study like those of Pecina [22] and Sikora [23], designed specifically to study the mechanism of antidepressant placebo response. Indeed, addressing the paucity of such targeted interrogation likely offers the best opportunity to advance antidepressant placebo response prophylaxis.

## Figures and Tables

**Figure 1 pharmaceuticals-12-00158-f001:**
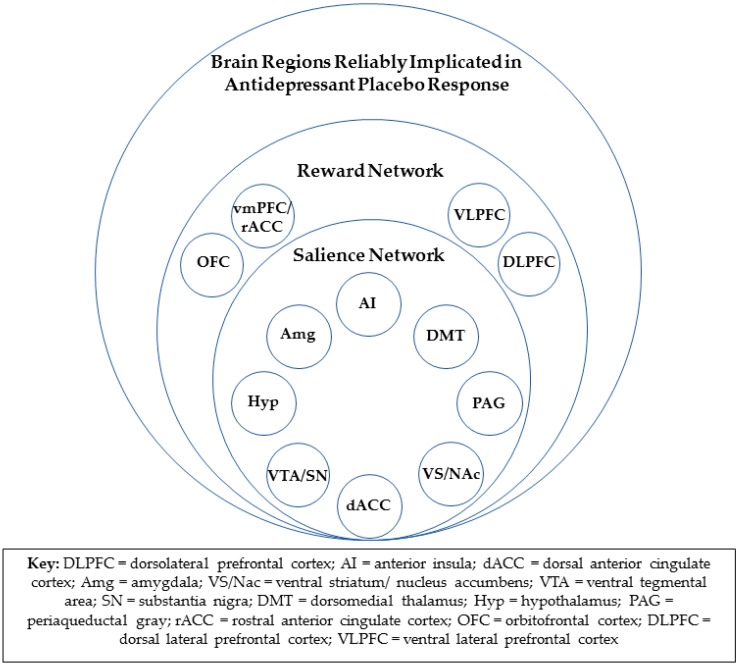
Brain regions reliably implicated in antidepressant placebo response and their associated networks.

**Table 1 pharmaceuticals-12-00158-t001:** Imaging data of the antidepressant placebo response, captured in clinical trials primarily focused on antidepressant efficacy or predicting antidepressant response.

Reference	N	Method	Brain Regions Associated with Antidepressant Placebo Response
[15]	17	FDG PET after 1 and 6 wks PBO or ADT	rACC, pACC, NAc, OFC, dlPFC, PMC, IPC, Hyp, Tha, SSA, pIns, pHipp; also predict ADT response.
[16,17]	51	EEG at baseline, after 1 week single-blind PBO lead-in, and 2, 4, 6, and 8 wks ADT	Frontocentral cortex
[18]	72	EEG at baseline	None
[19]	144	EEG at baseline, after 1 week PBO	rACC;
[20]	248	EEG at baseline, after 1 week PBO or ADT	rACC; also predicts ADT response
[21]	238	EEG at baseline, after 1 week PBO or ADT, functional connectivity analysis.	rACC, rAI; also predicts ADT response

Key: FDG = fluorodeoxyglucose; PET = positron emission tomography; PBO = placebo; ADT = antidepressant treatment; EEG = electroencephalogram; PFC = prefrontal cortex; PMC = premotor cortex; IPC = inferior parietal cortex; pIns = posterior insula; pACC = posterior anterior cingulate cortex; sgACC = subgenual anterior cingulate cortex; rACC = rostral anterior cingulate cortex; Hyp = hypothalamus; Tha = thalamus; SSA = supplementary sensory area; pHipp = parahippocampus.

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
