# Peer review of "Neural Predictors of the Antidepressant Placebo Response"

_pharmaceuticals, 2019, doi:10.3390/ph12040158_

Round 1

Reviewer 1 Report

The manuscript “Neural Predictors of Antidepressant Placebo Response” reviews available evidence of imaging correlates of placebo response with the aim of aiding drug development. The review is interesting and clear. My only suggestion is to include references in section 4.1

Author Response

Response to Reviewer 1 Comments

Point 1: The manuscript “Neural Predictors of Antidepressant Placebo Response” reviews available evidence of imaging correlates of placebo response with the aim of aiding drug development. The review is interesting and clear. My only suggestion is to include references in section 4.1.

Response 1: Thank you for your feedback. We appreciate your suggestion to add references to section 4.1. We have made this change and added references for the two studies mentioned in this section. The change has been made in line 224 in the manuscript. The change reads, “Two studies demonstrate unequivocally that endogenous opioid neurotransmission is involved in antidepressant placebo response [22, 23].”

The references include:

Peciña, M., Bohnert, A. S. B., Sikora, M., Avery, E. T., Langenecker, S. A., Mickey, B. J., & Zubieta, J.-K. (2015). Association between placebo-activated neural systems are linked to antidepressant responses: Neurochemistry of placebo effects in major depression. JAMA Psychiatry, 72(11), 1087–1094. https://doi.org/10.1001/jamapsychiatry.2015.1335

Sikora, M., Heffernan, J., Avery, E. T., Mickey, B. J., Zubieta, J.-K., & Peciña, M. (2016). Salience network functional connectivity predicts placebo effects in major depression. Biol Psychiatry Cogn Neurosci Neuroimaging, 1(1), 68–76. https://doi.org/10.1016/j.bpsc.2015.10.002

Reviewer 2 Report

Overall a very sound review of placebo effect for antidepressants.  The review presents many types of data and varying viewpoints.  Figure 1 was very helpful, as was Table 1.  While the manuscript text was concise and made excellent summarizations of literature, it would have been made even stronger if the authors could have performed some form of quantitative meta-analysis, especially given disparity in the field.  By meta-analysis, I mean combining cohort studies together to determine an aggregate effect and standard deviation of placebo across studies or comparing for different metrics, etc.  The only thing this paper lacked from getting 5-stars was the lack of quantitatively supported conclusion.  It does summarize qualitative data well, but a quantitative point, even if just a minor one, would have been more powerful.  Finally, the last paragraph could have been stronger by stating the authors' final point of view based on the review they completed.

Author Response

Response to Reviewer 2 Comments

Point 1: Overall a very sound review of placebo effect for antidepressants.  The review presents many types of data and varying viewpoints.  Figure 1 was very helpful, as was Table 1.  While the manuscript text was concise and made excellent summarizations of literature, it would have been made even stronger if the authors could have performed some form of quantitative meta-analysis, especially given disparity in the field.  By meta-analysis, I mean combining cohort studies together to determine an aggregate effect and standard deviation of placebo across studies or comparing for different metrics, etc.  The only thing this paper lacked from getting 5-stars was the lack of quantitatively supported conclusion.  It does summarize qualitative data well, but a quantitative point, even if just a minor one, would have been more powerful.  

Response 1: Thank you for your feedback. We appreciate your suggestion to add a quantitative meta-analysis and agree that including a quantitatively supported conclusion would advance the field. However, it is not within the scope of the current project. Instead we will look to do this in a future manuscript.

Point 2: Finally, the last paragraph could have been stronger by stating the authors' final point of view based on the review they completed.

Response 2: We agree and have modified the last two sentences (lines 297-301) to read as follows: “In a similar vein, examining the role dopamine neurotransmission may prove fruitful, especially in the context of a study like those of Pecina [22] and Sikora [23], designed specifically to study the mechanism of antidepressant placebo response. Indeed, addressing the paucity of such targeted interrogation likely offers the best opportunity to advance antidepressant placebo response prophylaxis.”